# The impact of early adjuvant chemotherapy in rectal cancer

Gyoung Tae Noh[1], Jeonghee Han[2], Min Soo Cho[3], Hyuk Hur[3], Kang Young Lee[3], Nam Kyu Kim[3], Byung Soh Min[3]*

1 Department of Surgery, Ewha Womans University College of Medicine, Seoul, South Korea, 2 Department of Surgery, Hallym University College of Medicine, Seoul, South Korea, 3 Department of Surgery, Yonsei University College of Medicine, Seoul, South Korea

* bsmin@yuhs.ac

## Abstract

### Purposes

Although adjuvant chemotherapy (AC) has been established as a standard of treatment for advanced rectal cancer, there is no guideline regarding the timing of AC initiation. In this study, we aimed to evaluate the oncologic outcome of early AC initiation and clarify the ideal time to AC among rectal cancer patients receiving preoperative chemo-radiotherapy (preCRT).

### Methods

The medical records of 719 patients who underwent curative resection followed by AC for rectal cancer were analyzed retrospectively. Data distributions were compared according to the calculated cut-off for AC initiation, survival results, and chemotherapy-induced toxicity. Additionally, patients were divided into two groups according to preCRT status and compared with respect to differences in the optimal time to AC.

### Results

Overall, a cut-off time point of 20 days after surgery for AC initiation was identified as the optimal interval; this yielded a significant difference in disease-free survival but no significant difference in AC toxicity. In the cut-off analysis of patients treated without preCRT, 19 days was identified as the optimal time to AC. However, for patients treated with preCRT, no significant value affected the survival outcome.

### Conclusions

Earlier initiation of AC (within approximately 3 weeks) was associated with better oncological outcomes among patients with rectal cancer. Additionally, the optimal timing of AC was unclear among patients who received preCRT; this might be attributable to an undetermined role of AC after preCRT or the effects of complications such as anastomotic leakage.

**Data Availability Statement:** All relevant data are within the manuscript and its Supporting Information files.

**Funding:** The authors received no specific funding for this work.

**Competing interests:** The authors have declared that no competing interests exist.

**Abbreviations:** AC, Adjuvant chemotherapy; ASA grade, American Society of Anesthesiologists grade; CEA, carcinoembryonic antigen; CECAE, Common Terminology Criteria for Adverse Events; CRC, Colorectal cancer; DFS, disease-free survival; ERAS protocol, Enhanced recovery after surgery protocol; MIS, minimally invasive surgery; OS, overall survival; pCR, pathologic complete response; preCRT, Preoperative chemo-radiotherapy; Q, quartile; SD, standard deviations; TNM system, tumor-node-metastasis system.

## Introduction

Adjuvant chemotherapy (AC) has been established as a current standard of treatment for colorectal cancer (CRC), although surgical resection is the primary treatment modality. Multiple trials of AC after curative resection for CRC have consistently demonstrated improvements in survival, and current guidelines recommend AC for patients with advanced CRC.[1–3] However, no guideline has been set regarding the timely initiation of AC, although the routine clinical assumption suggests that AC should be initiated as soon as possible.[2, 4, 5] A review of the literature indicates that early initiation of AC is most often defined as initiation within 8 weeks after surgery and has been shown to improve prognosis.[6–9] Therefore, a consensus has been reached regarding AC initiation within 8 weeks after curative surgery for CRC.[10–16] In recent decades, the application of enhanced recovery after surgery (ERAS) protocols and development of minimally invasive surgery (MIS) techniques have enabled earlier initiation of postoperative chemotherapy. ERAS protocols and MIS techniques such as laparoscopic surgery yield shortened hospital stays and more rapid postoperative recovery relative to conventional practices, and subsequently allow earlier initiation of AC.[17–27] Whether AC initiation earlier than 8 weeks can provide further improvements, however, is yet to be established.

In contrast to colon cancer, current guidelines recommend preoperative chemo-radiotherapy (preCRT) for patients with advanced rectal cancer.[3, 5] Previous studies evaluating the optimal timing of AC specifically in rectal cancer suggested a period of 8 weeks after surgery, similar to colon cancer; however, preCRT was not considered.[28, 29] Currently, there is insufficient evidence to support to AC initiation within 8 weeks after surgery for patients who received preCRT.

This study aimed to evaluate the oncologic outcomes of early AC initiation and clarify the ideal time to AC among patients with rectal cancer. We also sought to investigate the difference in the optimal time to AC initiation between patients treated with and without preCRT.

## Materials and methods

The medical records of consecutive patients who underwent curative resection and received AC for the treatment of rectal cancer from January 2006 to December 2012 were reviewed retrospectively. The study was reviewed and approved by the Severance Hospital Institutional Review Board. (IRB No. 4-2016-1007) A waiver of informed consent was approved by the Institutional Review Board given the retrospective nature of the study.

### Inclusion and exclusion criteria

The eligibility criteria were a histologically confirmed rectal adenocarcinoma located within 15 cm from the anal verge, and major rectal resection with curative intent followed by AC. Patients who underwent R2 resection for macroscopic residual disease or non-resectional procedures for rectal cancer and those who did not receive AC were excluded.

### Treatment protocol

**Patients treated without preoperative chemo-radiation therapy.** Standard total mesorectal excision procedures were performed. AC was performed for patients with pathologic stage II and more by medical oncologists using individualized initiation timing plans that considered patients' postoperative recovery. AC regimens were categorized and analyzed according to base chemotherapeutic agents, rather than according to delivery methods. Selected patients received postoperative radiotherapy with respect to the tumor location, tumor invasion depth, perirectal lymph node metastasis, circumferential resection margin, comorbidities,

and postoperative performance scale. All surgical patients were followed up at 3- or 6-month intervals for the first 5 years and annually thereafter.

**Patients with preoperative chemo-radiation therapy.** PreCRT was performed as long-course radiotherapy (50.4 Gy radiation/28 fractions/6 weeks) with 2 concurrent cycles of 5-fluorouracil (5-FU) infusion or oral capecitabine. Definite surgery was performed 4–8 weeks after the termination of CRT. AC was performed for patients with initial clinical stage II and more at diagnosis. Surgical maneuvers, AC, and follow-up were identical to those described for patients treated without preCRT. Postoperative radiotherapy was not indicated for these patients.

## Variables and outcomes

A survival analysis was performed to identify the associations between the time to AC and oncologic outcomes; the following variables were included in the analysis: age, sex, American Society of Anesthesiologists (ASA) grade, preoperative carcinoembryonic antigen (CEA) level, preCRT, surgical method (open surgery or MIS), pathological stage, histologic grade, lympho-vascular invasion, anastomotic leakage, duration of hospital stay after surgery, time to AC, and chemotherapy regimen. MIS included laparoscopic and robotic surgery. Pathologic stage was based on the seventh edition of the American Joint Commission on Cancer tumor-node-metastasis (TNM) system and included both pTNM for patients treated without preCRT and ypTNM for patients treated with preCRT.[30] Anastomotic leakage was defined as the break-down of a colorectal anastomosis along with infected fluid collection in the pelvic cavity; this condition was diagnosed by using computed tomography findings or clinical symptoms and signs, including a change in drainage color and/or fever with peritonitis. Time to AC was defined as the number of days between curative rectal cancer surgery and the initial chemo-therapeutic agent administration. We adopted disease-free survival (DFS) and overall survival (OS) as oncologic outcomes. Patients who experienced grade ≥3 chemotherapy-induced com-plications and required chemotherapy dose reductions or discontinuation because of toxicity were analyzed to identify the safety of and tolerance to early AC initiation. In this analysis, complication grades was based on the Common Terminology Criteria for Adverse Events (CTCAE), version 4.0.[31] To investigate differences in the optimal time of AC initiation between patients treated with and without preCRT, we categorized patients into two groups based on preCRT status and analyzed them, respectively.

## Statistical analysis

All statistical analyses were performed using SPSS Statistics (version 20.0.; IBM Corp., Armonk, NY, USA), except calculations of the time to AC initiation cut-off point. Descriptive results are presented as medians with interquartile ranges (quartile [Q]1–Q3) for continuous outcomes and as frequencies and percentages for categorical outcomes. Differences in survival between patients who initiated AC within and beyond the cut-off point were estimated using the Kaplan–Meier method and compared using the log-rank test. Factors associated with DFS and OS were analyzed using a Cox proportional hazards regression analysis. In this analysis, the continuous variables of CEA and time to AC were dichotomized based on the normal limi-tation of 5 ng/ml and the calculated cut-off point, respectively. The optimal cut-offs for the time to AC initiation were assessed via maximally selected log-rank statistics, using the R Max-stat package (version 3.2.2.; R Foundation for Statistical Computing, Vienna, Austria).[32] All variables in the risk set were assessed as putative prognostic factors for DFS and OS in an unadjusted Cox regression. Variables with a *P* value of <0.10 in the unadjusted Cox regression were selected for the risk-adjusted Cox regression. A binary logistic regression model was used

to identify the risks of chemotherapy-induced complications and tolerance to chemotherapy according to the time to AC initiation. A $P$ value of <0.05 was considered statistically significant.

## Results

### Patient characteristics

A total of 977 patients were included in this analysis and were followed up for a median of 52.0 months (range, 32.0–70.0 months). The median interval between surgery and AC was 27.0 days (range, 21.0–33.0 days). Among the total patients, 872 (89.3%) and 251 (25.7%) received AC within 6 and 3 weeks, respectively (Fig 1). The median age of the patients was 60.0 years (range, 52.0–67.0 years), and more men (63.9%) than women (36.1%) were included (Table 1). Among all patients, 258 (26.4%) underwent preCRT, whereas 719 (73.6%) did not. MIS was predominant (62.7%), compared to open surgery (37.3%). Most patients (83.2%) had pathologic stage II or III disease. Nine patients (0.9%) achieved a pathologic complete response (pCR, yp0) after preCRT, yielding a pCR rate of 3.5% (9/258 patients treated with preCRT). There were 111 patients (11.4%) presented with distant metastasis and underwent curative surgery. Sixty-eight patients (7.0%) experienced anastomotic leakage after surgery, and all such complications occurred before AC initiation. The median hospital stay duration after surgery was 10.0 days (range, 8.0–14.0 days). Regarding chemotherapeutic regimens, 5-FU only chemotherapy was most frequently used for AC (675 patients, 69.1%), followed by 5-FU plus

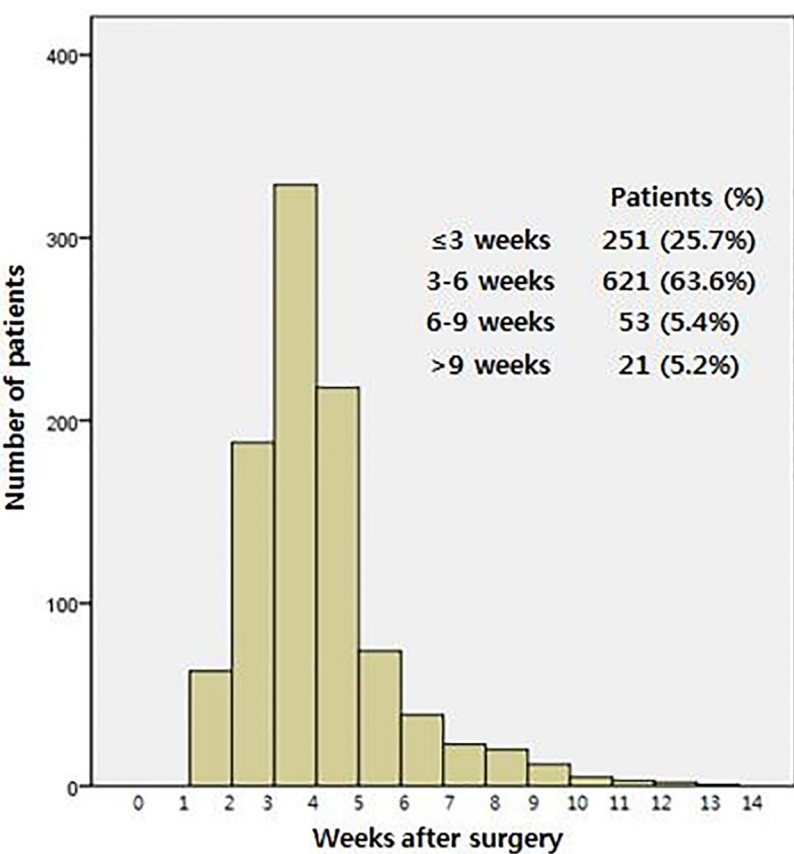

**Fig 1. Distribution of the time to adjuvant chemotherapy in 977 patients.**

**Table 1. Characteristics of all patients.**

| Demographics (N = 977) | |
|---|---|
| Age (years) | 60.0 (52.0–67.0) |
| Sex | |
| Male | 624 (63.9%) |
| Female | 353 (36.1%) |
| [†]ASA score | |
| I | 616 (63.1%) |
| II | 332 (34.0%) |
| III | 29 (3.0%) |
| Preoperative [‡]CEA (ng/ml) | |
| ≤5 ng/ml | 640 (65.5%) |
| >5 ng/ml | 312 (31.9%) |
| Unidentified | 25 (2.6%) |
| Preoperative chemo-radiotherapy | |
| Yes | 258 (26.4%) |
| No | 719 (73.6%) |
| Surgery method | |
| Open surgery | 364 (37.3%) |
| Minimally invasive surgery | 613 (62.7%) |
| Pathologic stage (p or yp) | |
| 0 | 9 (0.9%) |
| I | 44 (4.5%) |
| II | 338 (34.6%) |
| III | 475 (48.6%) |
| IV | 111 (11.4%) |
| Histologic grade | |
| I | 115 (11.8%) |
| II | 805 (82.4%) |
| III | 57 (5.8%) |
| Lymphovascular invasion | |
| Present | 678 (69.4%) |
| Absent | 299 (30.6%) |
| Anastomotic leakage | |
| Present | 68 (7.0%) |
| Absent | 909 (93.0%) |
| Postoperative hospital stay | 10.0 (8.0–14.0) |
| Time to adjuvant chemotherapy | 27.0 (21.0–33.0) |
| Regimen for adjuvant chemotherapy | |
| [Ÿ]5-FU only | 675 (69.1%) |
| 5-FU plus oxaliplatin | 208 (21.3%) |
| Capecitabine only | 77 (7.9%) |
| 5-FU plus Irinotecan | 17 (1.7%) |

Data are presented as medians (interquartile range, Q1-Q3), or n (%).

[†]ASA, American Society of Anesthesiologists;

[‡]CEA, carcinoembryonic antigen;

[Ÿ]5-FU, 5-fluorouracil

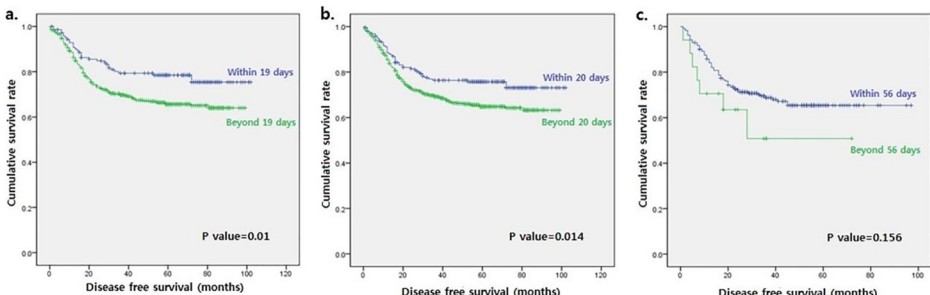

**Fig 2. Kaplan–Meier curves for disease-free survival among all the patients (a), patients treated without preoperative chemo-radiotherapy (b), and patients treated with preoperative chemo-radiotherapy (c).** Comparisons of patients who initiated adjuvant chemotherapy within the cut-off point (blue line) and beyond the cut-off point (green line).

oxaliplatin chemotherapy (208 patients, 21.3%). Capecitabine only and 5-FU plus irinotecan were used in 77 (7.9%) and 17 patients (1.7%), respectively.

## Analysis for overall patients

On calculation of the cut-off point of AC initiation among all the patients, 20 days after surgery was identified as the optimal interval with regard to DFS. A significant difference was observed when DFS was compared between patients who initiated AC within and beyond 20 days (Fig 2a). Patients who initiated chemotherapy within 20 days had a better 5-year DFS, compared to those who initiated chemotherapy beyond 20 days (75.8% vs. 64.8%, p = 0.014). However, the groups did not differ significantly in terms of OS (81.0% vs. 81.1%, respectively, p = 0.498). In the multivariate analysis of DFS, patients who received AC beyond 20 days had a significant hazard ratio (HR) of 1.5 when compared to patients who received AC within 20 days (95% confidence interval [CI]: 1.1–2.1; p = 0.01) after adjusting for potential confounders (Table 2).

Among all patients, 218 (22.3%) developed grade 3 or 4 chemotherapy-induced complications such as stomatitis, neutropenia, and diarrhea. A total of 226 patients (23.1%) required dose reduction, and the rate of chemotherapy discontinuation because of toxicity was 5.5% (n = 54). An analysis of safety and tolerance to early AC found no difference in the incidence of grade ≥3 chemotherapy-induced complications between the patients who initiated AC within and beyond 20 days after surgery (Table 3). Those in the latter group exhibited poorer tolerance to chemotherapeutic toxicity (odds ratio [OR] = 1.5; 95% CI: 1.0–2.12; p = 0.03). After adjusting for potential confounders, the result was not significant but demonstrated a trend towards increased risk. (OR = 1.4; 95% CI: 0.99–2.1; p = 0.06).

## Subgroup analysis of patients according to preoperative chemo-radiotherapy status

A cut-off point analysis for DFS among the patients treated without preCRT (median time to AC = 26.0 days [range, 21.0–32.0 days]) identified 19 days as the optimal time to AC. Patients who initiated AC within 19 days had a better DFS, compared to those who initiated AC beyond 19 days (p = 0.01; Fig 2b). In a multivariate analysis for DFS, patients who initiated AC beyond 19 days had a significant HR of 1.7, compared to those who initiated AC within 19 days (95% CI: 1.1–2.5; p = 0.01) after adjusting for potential confounders. Additionally, post-operative radiotherapy, which was performed in 328 (45.6%) of the 719 patients treated without preCRT, was not relevant to DFS and OS in this analysis. Among patients treated with preCRT (median time to AC = 31.0 days [range, 25.8–37.3 days]), no significant value affected

**Table 2. Cox regression disease-free survival in all patients.**

| Variables | Univariate analysis | | Multivariate analysis | |
|---|---|---|---|---|
| | HR (95% CI) | *P* | HR (95% CI) | *P* |
| Age | 1.000 (0.989–1.011) | 0.940 | | |
| Sex (female) | 0.736 (0.574–0.942) | 0.015 | 0.667 (0.518–0.858) | 0.002 |
| [†]ASA | 1.113 (0.907–1.365) | 0.306 | | |
| Surgical method | | 0.004 | | 0.314 |
| Open surgery | 1 | | 1 | |
| Minimally invasive surgery | 0.717 (0.570–0.902) | | 0.883 (0.692–1.126) | |
| [‡]CEA (>5 ng/ml) | 1.862 (1.478–2.347) | <0.001 | 1.394 (1.098–1.771) | 0.006 |
| Pathologic stage | 2.647 (2.233–3.137) | <0.001 | 2.183 (1.768–2.696) | <0.001 |
| Histologic grade | 1.755 (1.322–2.331) | <0.001 | 1.459 (1.089–1.939) | 0.011 |
| Lymphovascular invasion (present) | 2.017 (1.604–2.536) | <0.001 | 1.329 (1.042–1.694) | 0.022 |
| Anastomotic leakage (present) | 1.382 (0.910–2.099) | 0.129 | | |
| Time to adjuvant chemotherapy (>20 days) | 1.471 (1.077–2.010) | 0.015 | 1.520 (1.102–2.097) | 0.011 |
| Postoperative hospital stay | 1.024 (1.012–1.037) | <0.001 | 1.016 (1.003–1.029) | 0.015 |
| Chemotherapy regimen | | <0.001 | | 0.647 |
| [ϒ]5-FU only | 1 | | 1 | |
| 5-FU plus oxaliplatin | 1.001 (0.623–1.607) | | 1.201 (0.745–1.935) | |
| Capecitabine only | 2.185 (1.700–2.809) | | 1.130 (0.842–1.516) | |
| 5-FU plus Irinotecan | 4.766 (2.708–8.388) | | 1.390 (0.728–2.635) | |

[†]ASA, American Society of Anesthesiologists;

[‡]CEA, carcinoembryonic antigen;

[ϒ]5-FU, 5-fluorouracil.

the survival outcome. Patients who underwent AC within 56 days after surgery tended to have a better DFS, compared to other patients (p = 0.156; Fig 2c). No significant difference in OS was observed with respect to the cut-off point in either subgroup.

## Factors preventing early adjuvant chemotherapy

The major factors that prevented early AC (i.e., before the defined cut-off point) were general weakness (objective patient condition deemed inadequate for AC according to clinicians' judgment), poor compliance (patients' own refusal despite clinicians' recommendation), and anastomosis-related complications. (Table 4) The influencing factors differed with respect to

**Table 3. Odds ratios for patients who initiated chemotherapy beyond 20 days relative to within 20 days after surgery.**

| | Odds ratio (95% confidence interval) | *P* |
|---|---|---|
| Chemotherapy-induced complication grade ≥3 | | |
| Not adjusted | 0.921 (0.635–1.335) | 0.664 |
| Adjusted for age, [†]ASA, and chemotherapeutic regimen | 0.895 (0.611–1.312) | 0.571 |
| Dose reduction or discontinuation of chemotherapy due to toxicity | | |
| Not adjusted | 1.504 (1.037–2.180) | 0.031 |
| Adjusted for age, ASA, and chemotherapeutic regimen | 1.444 (0.987–2.112) | 0.058 |

[†]ASA, American Society of Anesthesiologists.

Table 4. **Factors preventing early postoperative chemotherapy.**

| | Total patients (N = 593) | Preoperative chemo-radiotherapy (-) (N = 576) | Preoperative chemo-radiotherapy (+) (N = 17) |
|---|---|---|---|
| General weakness | 241 (40.6%) | 237 (41.1%) | 4 (23.5%) |
| Poor compliance | 223 (37.6%) | 223 (38.7%) | |
| †Anastomosis related complications | 66 (11.1%) | 56 (9.7%) | 10 (47.1%) |
| Postoperative ileus | 23 (3.9%) | 23 (4.0%) | |
| ‡Infectious complications | 13 (2.2%) | 13 (2.3%) | |
| ÿCardiovascular complications | 6 (1.0%) | 5 (0.9%) | 1 (5.9%) |
| Others | 21 (3.5%) | *19 (5.7%) | **2 (29.4%) |

†Anastomosis related complications: anastomotic leakage, anastomotic bleeding, anastomotic ulcer

‡ Infectious complications: wound infection, pneumonia, pseudomembranous colitis perianal abscess, acute pyelonephritis, fever of unknown origin

ÿCardiovascular complications: Ischemic heart disease, atrial fibrillation, pulmonary embolism, cerebral infarction, ischemic colitis

*others for preoperative chemoradiotherapy (-): chyloperitoneum, acute renal failure, femoral neuropathy, acute appendicitis, burn, endometriosis, iatrogenic bowel injury, neurogenic bladder, aggravated underlying disease (liver cirrhosis, anxiety disorder, gastric ulcer)

**others for preoperative chemoradiotherapy (+): acute renal failure

treatment strategy (i.e., preCRT status). Among patients treated without preCRT, 576 underwent AC beyond the cut-off point of 19 days after surgery. Among them, 237 (41.1%) and 223 patients (38.7%) exhibited general weakness and poor compliance, respectively, two of the leading causes known to interfere with early AC among patients treated without preCRT. Anastomosis-related complications, particularly anastomotic leakage, were observed in 56 patients (9.7%). Among those treated with preCRT, 17 underwent AC beyond the cut-off point of 56 days after surgery. Ten of these patients (47.1%) presented with anastomosis-related complications, the leading cause of delayed AC. Among patients treated with preCRT, surgical complications such as anastomotic leakage were a major reason for delaying AC, in contrast to patients treated without preCRT.

## Discussion

Without a guideline for timely AC initiation, the routine clinical assumption is that AC should be initiated as soon as possible.[2, 4, 5] Theoretically, the early initiation of AC suggests a certain benefit. The Goldie–Coldman mathematical model predicts the probability of mutations that lead to increased drug resistance over time and depend on the mutation rate and tumor size.[33] Animal model studies suggest that surgery might increase the number of circulating tumor cells and potentiate the growth of metastatic deposits in response to the enhanced production of oncogenic growth factors after surgery.[34–39] This kinetic model and research results from preclinical tumor growth studies support the advantage of early AC initiation with respect to the eradication of micrometastatic deposits. Accordingly, the initiation of AC within 8 weeks after surgery has been traditionally recommended.[10–16]

Nowadays, the time to AC has shrunk following the application of ERAS protocols and expansion of MIS techniques. In our institution, the application of a "critical pathway" and expansion of MIS enabled the early initiation of AC; for example, 89.3% of patients in the present study received AC within 6 weeks. Such circumstances challenge the value of the previously recommended optimal timing of AC (i.e., within 8 weeks). A recent study of the oncologic outcomes of early AC initiation in colon cancer reported that patients who received AC within 3 weeks had a better DFS, compared to those who initiated AC beyond 3 weeks.[40] In the present study of rectal cancer, patients who initiated AC within 20 days had a significantly better DFS, compared to those who initiated AC beyond 20 days. Regarding the safety

of early AC, patients who started AC within and beyond 20 days did not differ in terms of the incidence of chemotherapy-induced complications.

The effect of AC on the prognosis of patients treated with preCRT remains debatable. Although the National Comprehensive Cancer Network and European Society for Medical Oncology guidelines recommend AC after preCRT and surgery, the Dutch and Norwegian guidelines do not recommend AC for patients who have received preCRT.[41] Furthermore, multiple randomized trials have reported a lack of oncologic benefit from AC among patients who received preCRT.[42–46] In the present study, we observed no significant cut-off point for AC initiation with regard to oncologic outcomes. We observed only a trend toward better survival among the patients who received AC within 8 weeks. Because anastomosis-related complications comprised the major cause of AC delays beyond 8 weeks (47.1%) and as anastomotic leakage is a well-known risk factor for cancer recurrence, the observed trend toward better oncologic outcomes might indicate a bias toward poor outcomes among patients with leakage.[47, 48]

Despite the limitations of this retrospective study, the results provide a comprehensive analysis of the optimal time for AC initiation among patients with rectal cancer, while considering how preCRT strategies for this type of cancer differ from those for colon cancer. In the era of MIS, which enables early recovery from surgery, these analyses may provide further insights into the adjuvant treatment of rectal cancer. Certainly, further studies will be required to establish the optimal timing of AC and facilitate the management of such patients.

## Conclusions

In conclusion, this study suggests that the earlier initiation of AC (i.e., within approximately 3 weeks) was associated with better oncological outcomes, especially DFS, among patients with rectal cancer. Additionally, among patients who received preCRT, the optimal timing of AC was unclear and could be attributed to the undetermined role of AC after preCRT or the effects of complications such as anastomotic leakage.

## Supporting information

**S1 File. Data file.**
(XLS)

## Author Contributions

**Conceptualization:** Gyoung Tae Noh, Byung Soh Min.

**Data curation:** Gyoung Tae Noh, Jeonghee Han, Min Soo Cho, Hyuk Hur, Kang Young Lee, Nam Kyu Kim, Byung Soh Min.

**Formal analysis:** Gyoung Tae Noh.

**Investigation:** Gyoung Tae Noh, Jeonghee Han, Byung Soh Min.

**Methodology:** Gyoung Tae Noh, Byung Soh Min.

**Resources:** Gyoung Tae Noh, Min Soo Cho, Hyuk Hur, Kang Young Lee, Nam Kyu Kim, Byung Soh Min.

**Supervision:** Byung Soh Min.

**Validation:** Gyoung Tae Noh, Byung Soh Min.

**Writing – original draft:** Gyoung Tae Noh.

**Writing – review & editing:** Gyoung Tae Noh, Min Soo Cho, Hyuk Hur, Kang Young Lee, Nam Kyu Kim, Byung Soh Min.

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
