## [Decision Letter · Decision Letter 0]

19 Sep 2019

PONE-D-19-23215

The impact of early adjuvant chemotherapy in rectal cancer

PLOS ONE

Dear Min,

Thank you for submitting your manuscript to PLOS ONE. After careful consideration, we feel that it has merit but does not fully meet PLOS ONE’s publication criteria as it currently stands. Therefore, we invite you to submit a revised version of the manuscript that addresses the points raised during the review process.

We would appreciate receiving your revised manuscript by Nov 03 2019 11:59PM. To enhance the reproducibility of your results, we recommend that if applicable you deposit your laboratory protocols in protocols.io, where a protocol can be assigned its own identifier (DOI) such that it can be cited independently in the future. For instructions see: http://journals.plos.org/plosone/s/submission-guidelines#loc-laboratory-protocols

We look forward to receiving your revised manuscript.

Kind regards,

Ju-Seog Lee

Academic Editor

PLOS ONE

Reviewers' comments:

Reviewer's Responses to Questions

**Comments to the Author**

1. Is the manuscript technically sound, and do the data support the conclusions?

Reviewer #1: Yes

Reviewer #2: Partly

2. Has the statistical analysis been performed appropriately and rigorously? 

Reviewer #1: Yes

Reviewer #2: N/A

3. Have the authors made all data underlying the findings in their manuscript fully available?

Reviewer #1: Yes

Reviewer #2: Yes

4. Is the manuscript presented in an intelligible fashion and written in standard English?

Reviewer #1: Yes

Reviewer #2: Yes

5. Review Comments to the Author

Reviewer #1: The authors demonstrated the impact of early adjuvant chemotherapy in rectal cancer. They concluded that earlier initiation of adjuvant chemotherapy within 3weeks was associated with better oncological outcomes in the patients without preCRT in rectal cancer. The authors should clarify some issues to make this paper clearer.

1. Confounding factors for the timing of adjuvant chemotherapy

As the authors described in the manuscript, the initiation of AC is decided by patients' general condition and successful recovery from the surgical procedure. Therefore, several confounding factors regarding demographic and peri-operative factors may affect the time of adjuvant chemotherapy, especially in survival analysis such as Cox proportional hazards models. The authors should evaluate this point more specifically. The linear regression for the timing of AC can be done with demographic and peri-operative factors as well as multicollinearity test between the timing of AC and confounding factors in Cox's model. The comparison of early and late AC group can be performed precisely after the statistical adjustment such as propensity score matching.

2. Estimation of best cut-off time point

The authors demonstrated the 19 days as an optimal cut-off in patients without preCRT. However, mix effects from several confounding factors may affect this result. The patient group without pre-CRT should be analyzed more precisely in the sub-categories adjusted from confounding factors. This will reinforce the statistical impact of the main findings in this paper.

3. Completion vs. Incompletion of adjuvant chemotherapy as a prognostic factor

Did the completion of AC have prognostic significance in your dataset? Dose reduction or stopping of AC affect the survival outcome of the patients? Are there modifiable peri-operative factors that can affect the completion of AC? I think this can be an interesting point.

Reviewer #2: The authors analyzed 977 patients who had rectal cancer surgery, followed by adjuvant chemotherapy. They concluded that patients who received adjuvant chemotherapy within 20 days have a better DFS (not OS benefit), compared with those who received chemotherapy after 20 days.

I have several major concerns, and this is why I recommend major revision - this means that, once the concerns are properly addressed by the authors, the paper could be re-considered for publication.

1. At least in the US, where I practice medical oncology, neoadjuvant chemoRT (here, preCRT) is considered for patients with clinical T3/T4 or T1/T2 with involved lymph nodes. The authors specified the number of patients who received preCRT (258 pts, 26.4%) and those who did not. In table 1, there is no information about clinical (before surgery, not pathologic) T, or N staging, based on which neoadjuvant treatment is decided.

2. stage II + stage III = 34.6% + 48.6% = 83.2%, a majority of patients are stage II and III. As mentioned above, these patients usually receive preCRT. Why is that patients who received preCRT are only 26.4%, although stage II and III is 83%? Is it because patients were treated from Jan 2006 to Dec 2012, long time ago? If this is the case, the conclusion of this paper cannot be generalized to the current standard of care.

3. Most stage II and III patients receive preCRT, and therefore, your conclusion that benefit from early AC for those who received preCRT is not clear cannot be applied to the current standard of care.

4. Another major issue is chemotherapy mentioned in table 1. 5-FU 69.1%, oxaliplatin based 21.3%, capecitabine 7.9%, and irinotecan 1.7%. This sounds very unfamiliar to me. The stand of care for patients who did not receive neoadjuvant therapy, especially at least stage II or III, is adjuvant chemoradiation therapy and chemotherapy, not chemotherapy alone. Did your patients receive adjuvant chemotherapy alone without chemoRT? There is no information about this in the paper.

5. In addition to above, I don't know what they mean by 5-FU based?? do you mean 5-FU alone? oxaliplatin based? Do you mean oxaliplatin alone? If patients receive adjuvant chemotherapy, they usually receive FOLFOX, which means 5-FU plus oxaliplatin. What do you mean 5-FU or oxaliplatin based? What is capecitabine based? Is it capecitabine alone, or CAPOX? What is irinotecan-based? Do you mean irinotecan alone, or FOLFIRI, which is 5-FU plus irinotecan?

Based on above points, your analysis needs to be modified. I strongly recommend that you involve medical oncologists, given here all the authors are surgeons, or at least a few medical oncologists go over the manuscript before submission.

A few minor points

1. in table 1, why did stage IV patients receive surgery? It is 11.4%.

2. what is the "critical pathway" in your hospital? Is it similar to ERAS? Please briefly explain this.

6. PLOS authors have the option to publish the peer review history of their article (what does this mean?). If published, this will include your full peer review and any attached files.

Reviewer #1: No

Reviewer #2: No

---

## [Author Response · Author response to Decision Letter 0]

29 Oct 2019

Reviewer #1: The authors demonstrated the impact of early adjuvant chemotherapy in rectal cancer. They concluded that earlier initiation of adjuvant chemotherapy within 3 weeks was associated with better oncological outcomes in the patients without preCRT in rectal cancer. The authors should clarify some issues to make this paper clearer.

1. Confounding factors for the timing of adjuvant chemotherapy

As the authors described in the manuscript, the initiation of AC is decided by patients' general condition and successful recovery from the surgical procedure. Therefore, several confounding factors regarding demographic and peri-operative factors may affect the time of adjuvant chemotherapy, especially in survival analysis such as Cox proportional hazards models. The authors should evaluate this point more specifically. The linear regression for the timing of AC can be done with demographic and peri-operative factors as well as multicollinearity test between the timing of AC and confounding factors in Cox's model. The comparison of early and late AC group can be performed precisely after the statistical adjustment such as propensity score matching.

Answer) Given the retrospective nature of this study, accessible data estimating patients’ general condition and recovery were limited. Further investigation regarding the detailed factors will be required.

2. Estimation of best cut-off time point

The authors demonstrated the 19 days as an optimal cut-off in patients without preCRT. However, mix effects from several confounding factors may affect this result. The patient group without pre-CRT should be analyzed more precisely in the sub-categories adjusted from confounding factors. This will reinforce the statistical impact of the main findings in this paper.

Answer) The cut-point of adjuvant chemotherapy were assessed via maximally selected log-rank statistics, which traces the point that maximize the difference of the survival outcome between two groups without considering other factors. In this scaled analysis of 719 patients without preCRT, the result might be generalized without considering various confounding factors. Further study with larger population considering various confounding factors will be required.

3. Completion vs. Incompletion of adjuvant chemotherapy as a prognostic factor

Did the completion of AC have prognostic significance in your dataset? Dose reduction or stopping of AC affect the survival outcome of the patients? Are there modifiable peri-operative factors that can affect the completion of AC? I think this can be an interesting point.

Answer) We totally agree with your opinion. However, only 54 patients (5.5% of total patients) were underwent dose reduction or discontinuation of chemotherapy, which could not affect the result of current analysis. Furthermore, the timing of dose reduction or discontinuation of chemotherapy was heterogeneous. For these reasons, we did not include these factors in the analysis for survival. Further enlarged data set with increased patients underwent dose reduction or discontinuation of chemotherapy will be available for the analysis according to your opinion.

Reviewer #2: The authors analyzed 977 patients who had rectal cancer surgery, followed by adjuvant chemotherapy. They concluded that patients who received adjuvant chemotherapy within 20 days have a better DFS (not OS benefit), compared with those who received chemotherapy after 20 days.

1. At least in the US, where I practice medical oncology, neoadjuvant chemoRT (here, preCRT) is considered for patients with clinical T3/T4 or T1/T2 with involved lymph nodes. The authors specified the number of patients who received preCRT (258 pts, 26.4%) and those who did not. In table 1, there is no information about clinical (before surgery, not pathologic) T, or N staging, based on which neoadjuvant treatment is decided.

2. stage II + stage III = 34.6% + 48.6% = 83.2%, a majority of patients are stage II and III. As mentioned above, these patients usually receive preCRT. Why is that patients who received preCRT are only 26.4%, although stage II and III is 83%? Is it because patients were treated from Jan 2006 to Dec 2012, long time ago? If this is the case, the conclusion of this paper cannot be generalized to the current standard of care.

Answer) Neoadjuvant chemoRT was initiated in the late 1990’s as a treatment modality for rectal cancer. Validation of its efficacy in multiple trials of Dutch trial in 2001, CAO/ARO/AIO 94 in 2004, and Swedish study in 2005, et al., it has become the standard treatment for rectal cancer. In Korea, preCRT was adopted for rectal cancer treatment in 2004 and its usage has been increasing from 13.1% of rectal cancer in 2006 to 60% of rectal cancer in 2015. This retrospective study reviewed the patients’ records from 2006 to 2012. Because of the early phase of adoption of preCRT, there were many patients who did not underwent current standard treatment option and the records of clinical stage was not listed regularly. However, the criteria for adjuvant chemotherapy was obvious; Patients without preCRT: pathologic stage ≥ II, patient with preCRT: initial clinical stage ≥ II at diagnosis. We added these writing supplies in the manuscript. 

We think this study, which was not followed the current standard treatment strictly, may have value for some aspects. Nowadays, there are some concerns of overtreatment with radiotherapy because it caused several side effect and patients’ poor functional outcomes. So, some guidelines reduced the indication of radiotherapy than current guidelines. Even there are several ongoing trials ruling out radiotherapy for neoadjuvant treatment for rectal cancer. In this study, we categorized patients into two groups according to the experience of preCRT and analyzed it respectively. The results of each group may be of value for clinicians and clue for further study.

3. Most stage II and III patients receive preCRT, and therefore, your conclusion that benefit from early AC for those who received preCRT is not clear cannot be applied to the current standard of care.

Answer) We think current standard is not absolute. As we described in the manuscript, the effect of adjuvant chemotherapy on the prognosis of patients treated with preCRT remains debatable. Although NCCN and ESMO guidelines recommend AC after preCRT and surgery, The Dutch and Norwegian guidelines do not recommend AC for patients who have received preCRT. Furthermore, multiple randomized trials have reported a lack of oncologic benefit from adjuvant chemotherapy among patients who received preCRT. In this study, we observed no significant cut-off point for adjuvant chemotherapy initiation with regard to oncologic outcomes. We only suggested this finding can be associated with the lack of oncologic benefit of adjuvant chemotherapy for patients with preCRT, not conclusively.

4. Another major issue is chemotherapy mentioned in table 1. 5-FU 69.1%, oxaliplatin based 21.3%, capecitabine 7.9%, and irinotecan 1.7%. This sounds very unfamiliar to me. The stand of care for patients who did not receive neoadjuvant therapy, especially at least stage II or III, is adjuvant chemoradiation therapy and chemotherapy, not chemotherapy alone. Did your patients receive adjuvant chemotherapy alone without chemoRT? There is no information about this in the paper.

Answer) We categorized patients into two groups according to the experience of preCRT. In the manuscript, treatment protocol for each group was included. Selected patients without preCRT underwent postoperative radiotherapy with respect to the tumor location, invasion depth, perirectal lymph node metastasis, CRM, comorbidities, and postoperative performance scale. Postoperative radiotherapy was performed in 328 (45.6%) of the 719 patients without preCRT and this data is included in the manuscript.

5. In addition to above, I don't know what they mean by 5-FU based?? do you mean 5-FU alone? oxaliplatin based? Do you mean oxaliplatin alone? If patients receive adjuvant chemotherapy, they usually receive FOLFOX, which means 5-FU plus oxaliplatin. What do you mean 5-FU or oxaliplatin based? What is capecitabine based? Is it capecitabine alone, or CAPOX? What is irinotecan-based? Do you mean irinotecan alone, or FOLFIRI, which is 5-FU plus irinotecan?

Answer) That phrase can make a misunderstanding. We changed it more definite word in the manuscript and table.

A few minor points

1. in table 1, why did stage IV patients receive surgery? It is 11.4%.

Answer) We included stage IV patients with resectable metastasis, who underwent radical synchronous resection for both primary and metastatic lesion.

2. what is the "critical pathway" in your hospital? Is it similar to ERAS? Please briefly explain this.

Answer) We standardized a routine perioperative care protocol in 1999, and around 2008, we launched a modern perioperative patient care program called “Critical Pathway”, which is similar to ERAS but modified considering our medical circumstance.

---

## [Decision Letter · Decision Letter 1]

7 Jan 2020

The impact of early adjuvant chemotherapy in rectal cancer

PONE-D-19-23215R1

Dear Dr. Min,

We are pleased to inform you that your manuscript has been judged scientifically suitable for publication and will be formally accepted for publication once it complies with all outstanding technical requirements.

With kind regards,

Ju-Seog Lee

Academic Editor

PLOS ONE

Additional Editor Comments (optional):

Reviewers' comments:

Reviewer's Responses to Questions

**Comments to the Author**

1. If the authors have adequately addressed your comments raised in a previous round of review and you feel that this manuscript is now acceptable for publication, you may indicate that here to bypass the “Comments to the Author” section, enter your conflict of interest statement in the “Confidential to Editor” section, and submit your "Accept" recommendation.

Reviewer #1: All comments have been addressed

2. Is the manuscript technically sound, and do the data support the conclusions?

Reviewer #1: Yes

3. Has the statistical analysis been performed appropriately and rigorously? 

Reviewer #1: Yes

4. Have the authors made all data underlying the findings in their manuscript fully available?

Reviewer #1: Yes

5. Is the manuscript presented in an intelligible fashion and written in standard English?

Reviewer #1: Yes

6. Review Comments to the Author

Reviewer #1: (No Response)

7. PLOS authors have the option to publish the peer review history of their article (what does this mean?). If published, this will include your full peer review and any attached files.

Reviewer #1: Yes: Sung Hwan Lee

---

## [Editor Report · Acceptance letter]

23 Jan 2020

PONE-D-19-23215R1 

The impact of early adjuvant chemotherapy in rectal cancer 

Dear Dr. Min:

I am pleased to inform you that your manuscript has been deemed suitable for publication in PLOS ONE. Congratulations! Your manuscript is now with our production department. 

With kind regards,

on behalf of

Dr. Ju-Seog Lee 

Academic Editor

PLOS ONE